# Expanding chemical space by *para*-C−H arylation of arenes

Sudip Maiti[1], Yingzi Li[2], Sheuli Sasmal[1], Srimanta Guin[1], Trisha Bhattacharya[1], Goutam Kumar Lahiri[1✉], Robert S. Paton [2✉] & Debabrata Maiti [1,3✉]

Biaryl scaffolds are privileged templates used in the discovery and design of therapeutics with high affinity and specificity for a broad range of protein targets. Biaryls are found in the structures of therapeutics, including antibiotics, anti-inflammatory, analgesic, neurological and antihypertensive drugs. However, existing synthetic routes to biphenyls rely on traditional coupling approaches that require both arenes to be prefunctionalized with halides or pseudohalides with the desired regiochemistry. Therefore, the coupling of drug fragments may be challenging *via* conventional approaches. As an attractive alternative, directed C−H activation has the potential to be a versatile tool to form *para*-substituted biphenyl motifs selectively. However, existing C–H arylation protocols are not suitable for drug entities as they are hindered by catalyst deactivation by polar and delicate functionalities present alongside the instability of macrocyclic intermediates required for *para*-C−H activation. To address this challenge, we have developed a robust catalytic system that displays unique efficacy towards *para*-arylation of highly functionalized substrates such as drug entities, giving access to structurally diversified biaryl scaffolds. This diversification process provides access to an expanded chemical space for further exploration in drug discovery. Further, the applicability of the transformation is realized through the synthesis of drug molecules bearing a biphenyl fragment. Computational and experimental mechanistic studies further provide insight into the catalytic cycle operative in this versatile C−H arylation protocol.

[1] Department of Chemistry, Indian Institute of Technology Bombay, Mumbai 400076, India. [2] Department of Chemistry, Colorado State University, Fort Collins, CO 80523, USA. [3] IDP in Climate Studies, Indian Institute of Technology Bombay, 400076 Mumbai, India. ✉email: lahiri@chem.iitb.ac.in; robert.paton@colostate.edu; dmaiti@iitb.ac.in

Chemistries utilizing unactivated C–H bonds as potential sites for regioselective functionalization offer an efficient strategy for the decoration and diversification of biologically active molecules[1,2]. Alongside the growth of new directions in C–H transformations, there have been consistent efforts to harness broad functional group compatibility to expand the scope of these transformations to include complex molecules[3–5]. These advances have attracted the attention of the medicinal chemistry community, where C–H functionalization methods have been harnessed to explore chemical space inaccessible to conventional synthetic processes[6–8]. In this regard, C–H arylation to form biphenyl motifs can be an effective method of accessing areas of chemical space important for drug discovery and development[9]. This is attributed to the pivotal role of aromatic components in protein–ligand recognition[10] through non-covalent interactions, which can provide improved binding activity and selectvitiy[11–14]. In this context, a historical bias towards 'flatland' biphenyl products is usually observed with a *para*-connectivity[15]. Consequently, the development of a robust *para*-C–H arylation protocol to install biaryl units in complex drug molecules would allow rapid access to structural diversity and enhance the novel chemical space.

Druglike molecules present immediate challenges to the development of robust synthetic methodologies with transition metal catalysts. Drug moieties are commonly decorated with functionalities having Lewis basic heteroatoms[16–18], which tend to impede catalytic activity[19–21] or outcompete the directing group binding of the parent substrate[22]. As a result, efficient C–H activation protocols for the decoration and diversification of drugs are rare[23]. Nevertheless, the introduction of drug entities selectively at the *ortho*-position of arenes was recently demonstrated utilizing a directed C–H arylation protocol[24,25]. While *ortho*-substituted biaryls adopt non-planar conformations and experience restricted rotation, *para*-substitution results in flatland products associated with higher bioactivities[15]. In this endeavor, quite a few *para*-C–H arylation protocols of simple arenes have been developed by Gaunt[26], Yu[27], and Ye[28]. However, analogous directed C–H arylation protocols for availing such "flatland" biphenyls with *para*-connectivity remain elusive. Employing a transient mediator in substrates possessing a *meta*-directing group, *para*-arylation can be achieved[29,30]. However, the scope of these transformations is limited to 2,6-disubstituted arenes (Fig. 1a). This strategy is also not amenable to introducing heteroarene moieties, severely limiting its applicability in the preparation of pharmaceutically relevant fragments and drugs. Thus, the development of a general catalytic system is needed in which (i) the directing group (DG) outcompetes other Lewis basic atoms to bind the catalyst, and (ii) in which a stable macrocyclic intermediate required for DG-enabled *para*-C–H activation can be accessed[31–35].

Herein, we describe a robust catalytic system for conjugating arenes with drug molecules at the *para*-position, creating unique biphenyl fragments.

## Results

### Reaction optimization

Exploratory studies began by attempting the *para*-arylation of toluene derivative **1a** with 3-nitro iodobenzene. **1a** consists of a benzyl part tethered to an easily removable cyano containing biphenyl moiety (DG$_1$) by means of a silyl hinge (Fig. 1b). The incorporation of di-methoxy groups in DG$_1$ was found to be advantageous[36–39] (see Supplementary Information, Section 2.2 for further details). The *para*-arylation reaction initiated with Pd(OAc)$_2$ and *N*-Ac-Gly in the presence of AgOAc as oxidant provided the desired product. However, substantial improvement both in yield and selectivity was achieved

following further optimization of solvent and additives. Hexafluoroisopropanol (HFIP) solvent, which is beneficial in distal C–H activation, gave superior results to other solvents[40]. Although AgOAc showed promising results at early stages, extensive optimization studies through systematic alterations of different parameters led to the finding that a combination of Ag$_2$SO$_4$, Cu$_2$Cr$_2$O$_5$, and LiOAc.2H$_2$O, in appropriate proportion, gave the highest yield of *para*-arylated product. The regioselectivity could be further increased in favor of the desired *para*-arylated product by modifying the *N*-protecting group on the α-amino acid: Fmoc was found to enhance regioselectivity. An intriguing feature of these optimization studies was the influence of aryl iodide stoichiometry. A 1:1 ratio of aryl iodide to **1a** gave the maximum yield and selectivity under these conditions.

### Scope of the methodology

The optimized reaction conditions were then used to assess the potential of this protocol. **1a** was subjected to *para*-arylation with several aryl iodides, focusing on those with heteroatoms and functional groups found in drugs and natural products (Fig. 2a). Gratifyingly, the protocol was tolerant of an array of protected phenol derivatives, providing the desired *para*-arylated products in synthetically useful yields and selectivity, irrespective of aryl iodide regiochemistry (**3ab**-**3ag**; Fig. 2a). Similarly, aryl iodides possessing ester, sulfoxide, and amide groups all underwent facile reactions (**3ah**, **3ai**, **3aj**, and **3kt**; Fig. 2a). Functional groups vulnerable under oxidative conditions, *viz.* the formyl group, were also compatible under the present reaction conditions (**3ak** and **3ao**; Fig. 2a). The reaction with 4-iodo benzyl alcohol furnished the same product **3ak** (80%, *para*:others = 14:1) via in situ oxidation of the benzyl alcohol to a formyl group. Of significance are the high reactivities displayed by the *ortho*-substituted aryl iodides without compromising yields or selectivity despite the hindrance associated with their steric bulk (**3an**-**3bq**, Fig. 2a). The disubstituted aryl iodides with different substitution patterns were also competent substrates for *para*-arylation (**3ar** and **3as**, Fig. 2a). Typically, the regioselective heteroarylation of arenes is challenging to promote because competitive binding impedes catalytic activity. Remarkably, however, heteroarylation with different N, O, S heterocycles such as thiophene, furan, and quinoline derivatives was achieved in good yields and fairly high selectivities (**3au**-**3aw**, Fig. 2a). To the best of our knowledge, this is the first instance of the distal *para*-heteroarylation process via C($sp^2$)–H activation.

We next proceeded to examine the feasibility of *para*-arylation with a range of substituted arenes. An array of electronically diverse arenes possessing substituents at different positions were tolerated under the present reaction conditions (Fig. 2b). The electronic nature of the substituents did not significantly impact the yields of the corresponding *para*-arylated products. Functional groups at the *ortho*-position such as –Me, –OMe provided synthetically useful yields of their corresponding products with high *para*-selectivity (**3bm** and **3cx**, Fig. 2b). Different fluorine-containing substituents such as –CF$_3$ (**3dm**) and –SCF$_3$ (**3el**) were equally compatible in this transformation. Substituents at the *meta*-position such as –OMe was well tolerated with this reaction protocol (**3fm**, Fig. 2b). Reactive functional groups such as –Cl (**3hm**), prone to undergo cross-coupling with aryl iodides, remained intact. The presence of thiophene (**3iy**), biphenyl (**3jm**), and phenyl (**3km**) systems did not hamper the desired reaction. The presence of 2,6-, 2,5-, and 3,5-substituents, including fluorine atoms, led to *para*-arylated products in high yields and excellent selectivity (**3lm**-**3om**, Fig. 2b). The use of a naphthalene system provided C4-selectivity (**3pl** and **3pm**). The protocol was also applicable to toluene derivatives possessing α-substituents such as phenyl (**3ql** and **3rm**) and substituted phenyl (**3sz**). In the case of α-phenyl substrate, arylation occurred at one of

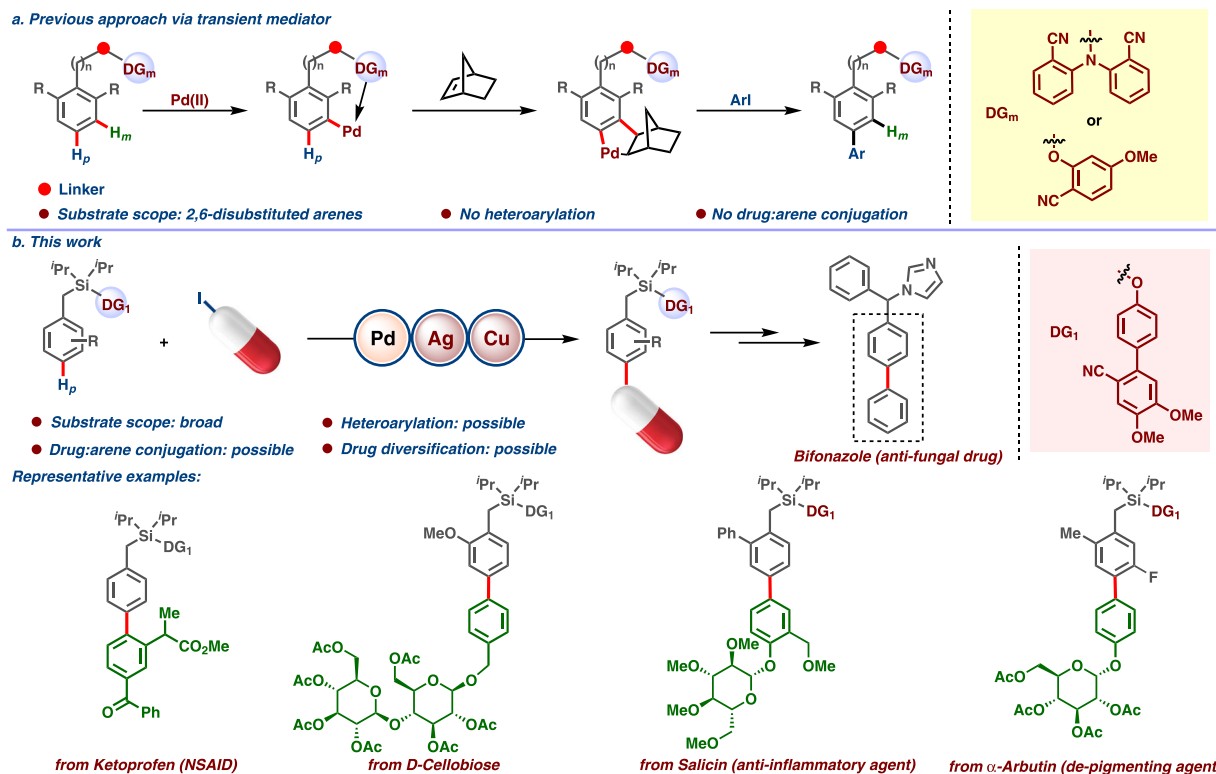

**Fig. 1 Expanding chemical space by *para*-C−H arylation. a** Previous work using palladium/norbornene cooperative catalysis. **b** Present work using traceless directing group.

the phenyl rings (**3ql** and **3rm**). In contrast to the above, the scope of *para*-arylation via a transient mediator is limited to 2,6-disubstituted arenes[29,30].

Encouraged by our protocol's compatibility with heteroatoms and heteroaryl groups, we next explored the potential for the selective incorporation of drugs and natural products at the *para*-position. Synthetic phenol *O*-glycosylated small molecules play a significant role as biochemical probes and therapeutic agents[41,42]. Their innate properties could be subtly modified by arylation of the phenol moiety of the glycoside[43]. To this aim, toluene derivatives were reacted with iodo-containing *O*-glycosylated phenols and iodo-benzyl alcohols (Fig. 3a). Strikingly, the presence of multiple heteroatoms in mono-saccharides such as D-glucose (**5aa**), D-galactose (**5ab**), D-mannose (**5ac** and **5tc**), L-rhamnose (**5ad**), D-xylose (**5qe**); di-saccharides such as D-maltose (**5kf**), D-lactose (**5kg, 5cg**, and **5sg**), and D-cellobiose (**5kh** and **5ch**) did not impede reactivity due to catalyst interception. High compatibility with such a wide range of hexoses and pentoses is rare in the entire realm of C−H arylation protocols. These successful outcomes to generate a library of structurally diversified *O*-aryl glycosides demonstrate the versatility of the protocol. Further, iodo-containing pharmaceuticals and agrochemicals could be employed as efficient coupling partners for *para*-C−H arylation of toluene derivatives (Fig. 3b). Saccharide-based drugs such as α-arbutin (**5ni**), β-arbutin (**5oj** and **5cj**), and salicin (**5kk, 5uk**, and **5sk**) were selectively introduced into *para*-position of arenes in useful yields and high selectivity. Similarly, ketoprofen (**5al**), marketed as nonsteroidal anti-inflammatory drugs (NSAIDs), was amenable to the reaction conditions to afford biaryl systems. These observations are of compelling interest to the field of drug diversification by C−H activation. Also, such a conjugation process enables an expanded chemical space with a desirable "flatland" and a modulated bioactivity in parent drugs.

**Application**. Removal of the directing group from the *para*-arylated product was achieved under mild conditions to afford the silanol (**6**, Fig. 4a). The potential of the silanol to act as an *ortho*-directing group was further exploited towards *ortho*-olefination (**7** and **8**, Fig. 4a). In an alternative approach, post *para*-arylation, treatment with tetrabutylammonium fluoride led to desilylation with the concomitant formation of *para*-arylated toluene and benzhydryl derivatives (**9** and **10**, Fig. 4b). Further, upon treatment with nitrosobenzene as an oxidant the C−Si bond of *para*-arylated product can be transformed into a valuable aldehyde (−CHO) functional group (**11**, Fig. 4c). Alternatively, the Fleming-Tamao oxidation of this C−Si bond of *para*-arylated product was able to deliver corresponding benzhydryl alcohol derivative, an important synthon for numerous drugs, natural products, and bio-active molecules (**12**, Fig. 4d)[44]. In the presence of TBAF the carbon atom of C−Si bond can also act as a potential nucleophile (Fig. 4e). The synthetic applicability was further elaborated by employing the *para*-arylated product to synthesize bifonazole, an antifungal drug (**15**, Fig. 4f). It is worth mentioning over here that the traditional method of synthesizing this drug molecule **15** uses Cl/Br-substituted benzophenone and expensive phenyl boronic acids which we are replacing by much cheaper iodo benzene in this protocol. The silyl linked DG can thus be viewed as a traceless DG that can be deleted or transformed into diverse functional groups.

**Mechanistic and computational studies**. Mechanistic studies were undertaken to gain insight into this transformation. A first-order dependence with respect to both the substrate and aryl iodide was observed, implying their involvement in or before the turnover-determining transition state (TDTS) (Fig. 5a). In addition, the absence of a primary kinetic isotopic effect (KIE) was

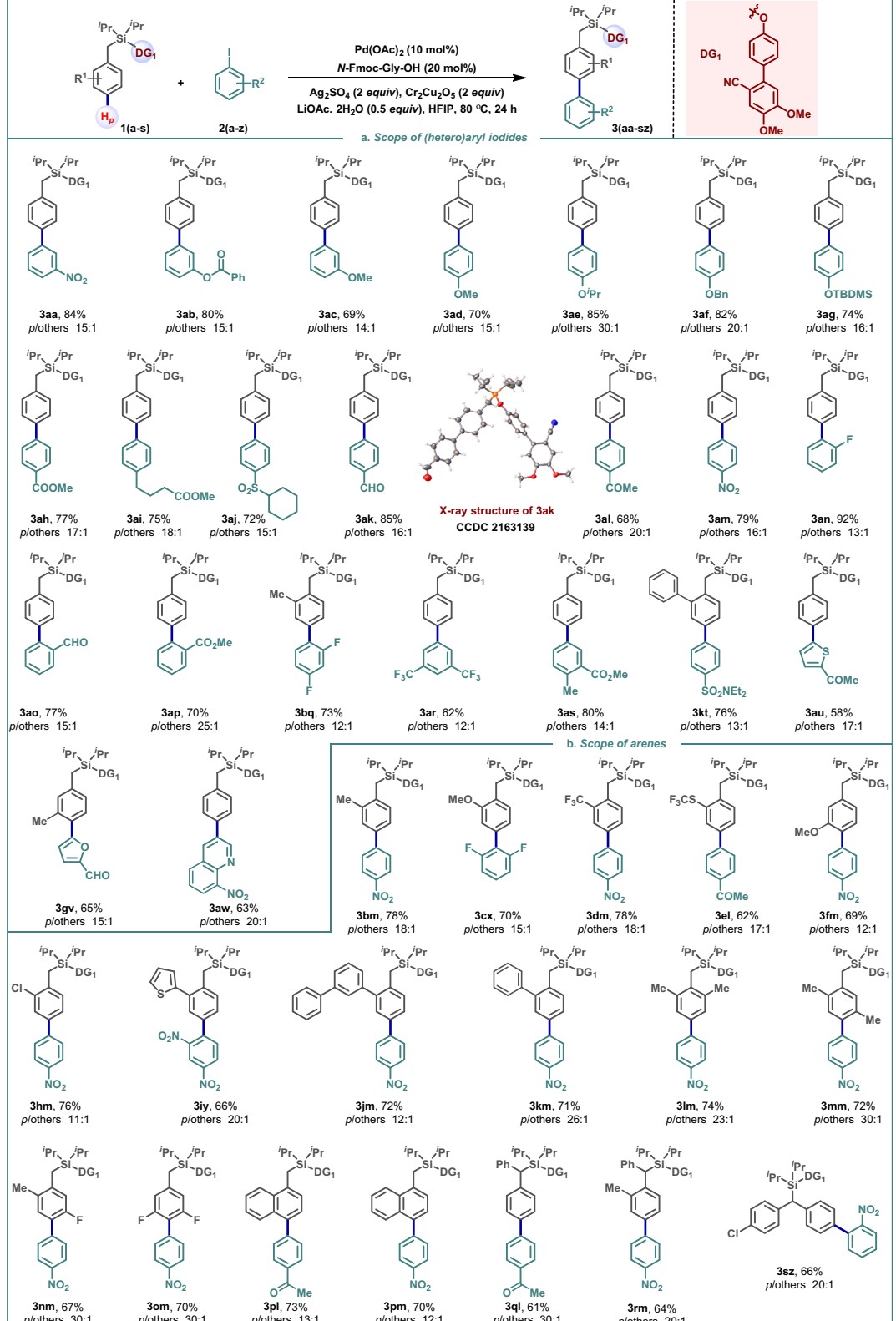

**Fig. 2 Scope of *para*-arylation. a** Scope of (hetero)aryl iodides. **b** Scope of substituted arenes. Reaction conditions: arene (1 equiv.), (hetero)aryl iodide (1 equiv.), Pd(OAc)$_2$ (10 mol%), *N*-Fmoc-Gly-OH (20 mol%), Ag$_2$SO$_4$ (2 equiv.), Cr$_2$Cu$_2$O$_5$ (2 equiv.), LiOAc.2H$_2$O (0.5 equiv.), HFIP, 80 °C, 24 h.

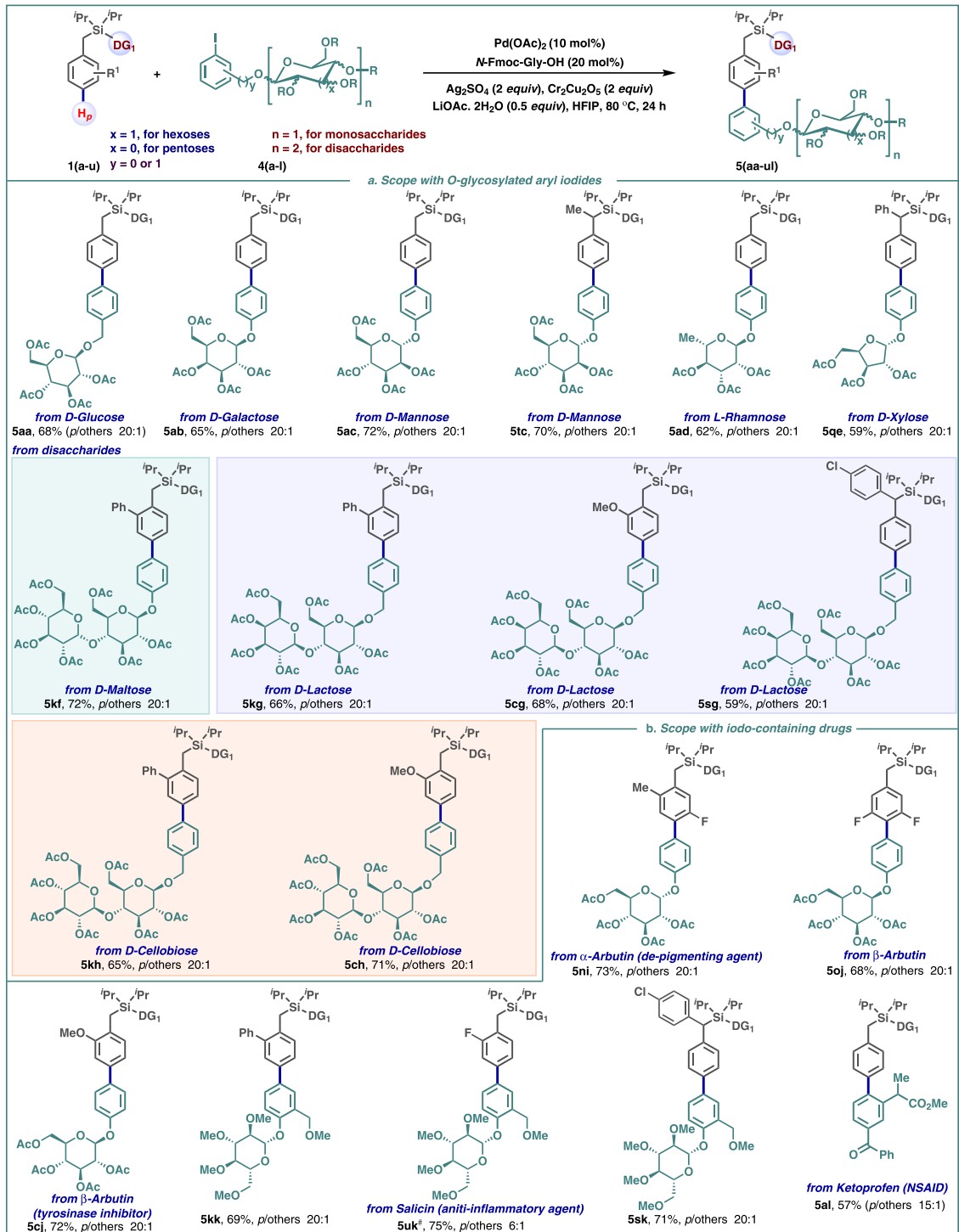

**Fig. 3 Scope of *para*-arylation. a** Scope of the reaction with *O*-glycosylated aryl iodides. **b** Scope of *para*-C–H arylation with iodo-containing drugs. Reaction conditions: arene (1 equiv.), (hetero)aryl iodide (1 equiv.), Pd(OAc)$_2$ (10 mol%), *N*-Fmoc-Gly-OH (20 mol%), Ag$_2$SO$_4$ (2 equiv.), Cr$_2$Cu$_2$O$_5$ (2 equiv.), LiOAc.2H$_2$O (0.5 equiv.), HFIP, 80 °C, 24 h. **#** the isomer for 5uk was not separated clearly.

observed from parallel experiments with protiated and deuterated substrates **1a** and **1a-d7** ($k_H/k_D = 1.03$), implying that C−H cleavage is not involved in the TDTS. Computational studies at the wB97XD/def2-TZVP//B3LYP/6−31G(d)+LANL08(f) level of theory with SMD solvation were used to obtain a Gibbs energy profile for the proposed catalytic cycle (Fig. 5b) (Calculations were performed with Gaussian 16 rev. B.01. Full computational details and references are given in the Supplementary

Information). Consistent with the experiment, these calculations predict C−H activation (**TS-1**) to be relatively facile compared to the subsequent oxidative addition of the aryl iodide (**TS-2**), which is the highest barrier overall. This is proceeded by reductive elimination (**TS-3**) to yield the biaryl product and regenerate the Pd(II) catalyst to continue the cycle. Mono- and multi-metallic mechanisms have been studied computationally in Pd-catalyzed C−H activations with mono-*N*-protected amino acid (MPAA)

**Fig. 4 Applications. a** DG removal and *ortho*-C−H functionalization. **b** DG removal with TBAF. **c** Oxidation of C−Si bond. **d** Fleming-Tamao oxidation of C−Si bond. **e** Nucleophilic attack of C−Si bond to aldehyde. **f** Formal synthesis of bifonazole.

ligands[36,45]. We considered both possibilities (see Supplementary Figs. 19, 20), finding that the mono-metallic pathway gave reasonable barriers which were also consistent with the observed regioselectivity.

Without the MPAA ligand, *para* C−H activation occurs via transition state **TS-3b** with an activation barrier of 15.7 kcal/mol, forming intermediate **int-4b** in an exergonic process. Subsequently, the oxidative addition of aryl iodide onto Pd(II) species via transition

state **TS-6b** with an energetic span of 35.7 kcal/mol leads to the formation of a Pd(IV) intermediate. This pathway would not occur under mild conditions. However, with the MPAA ligand, **TS-3a** contains a [5,6]-palladacycle conducive to C−H bond activation. The calculated free energy barrier of this step is 12.5 kcal/mol, in agreement with the absence of experimental KIE (measured $k_H/k_D = 1.03$). The oxidative addition of aryl iodide **2k** is computed to occur in the turnover frequency-determining transition state with a

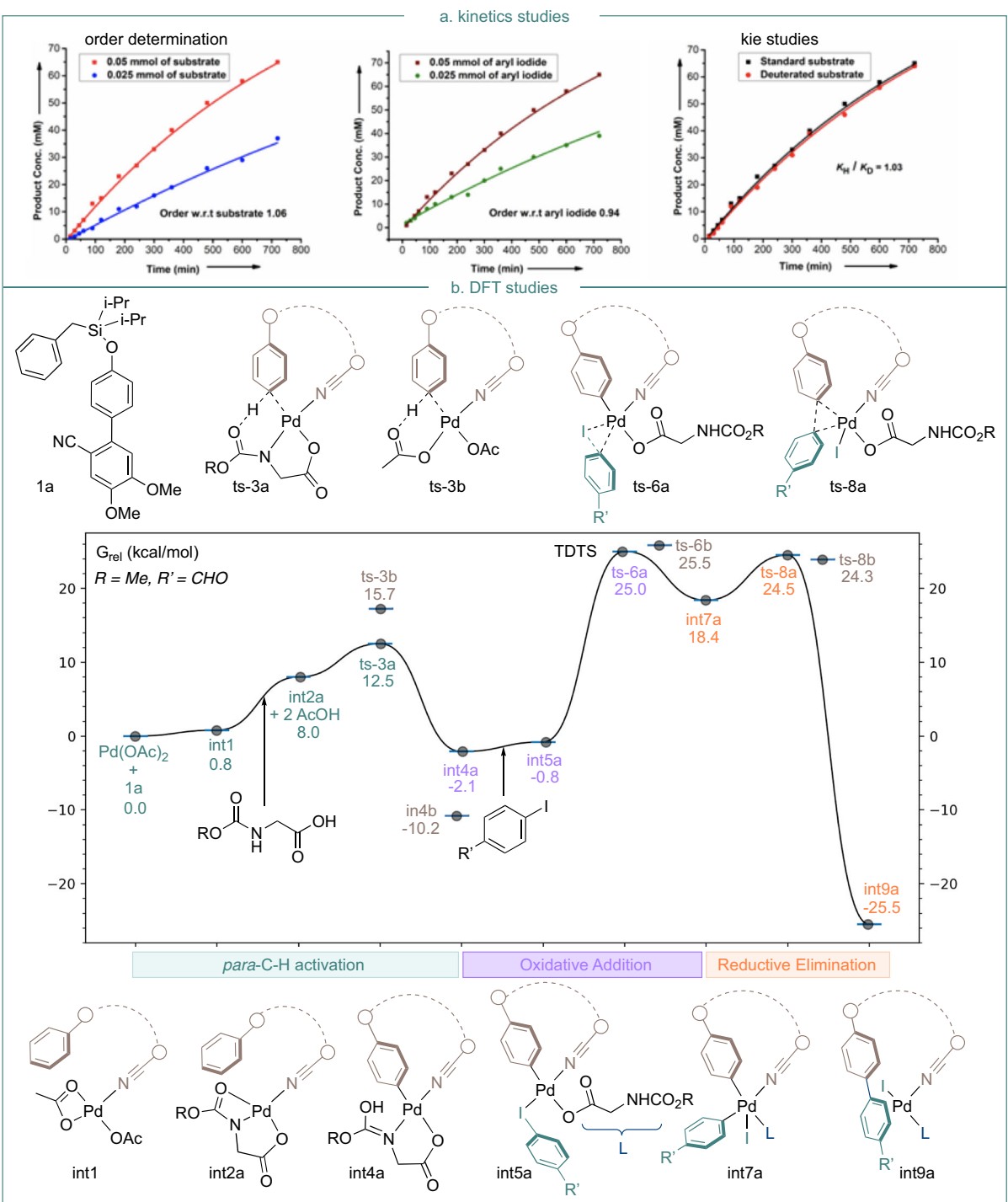

**Fig. 5 Mechanistic studies. a** Kinetic studies. **b** DFT studies [wB97XD/def2-TZVP//B3LYP/6-31G(d) + LANL08(f) computed Gibbs energy profile for the proposed mechanism of Pd-catalyzed *para*-C-H arylation reaction].

barrier of 27.1 kcal/mol from **int-6a**. Since C−H activation is reversible, it is this oxidative addition step that determines regioselectivity (*para* over *ortho* or *meta*) for the overall reaction. Consistent with the experiment, the *para*-**TS6a** is 2.5 and 3.7 kcal/mol more stable than the *meta*- and *ortho*-**TS6a** in Fig. 6a, respectively. The site selectivity results from differing ring strain in the palladacyclic TS, which is minimized in the *para* pathway. The relative macrocyclic ring strain energies in *ortho*, *meta*, and *para* TS structures were evaluated using isodemic reactions (Fig. 6a). These

studies suggest that relative to the 16-membered *para*-TS, the 15-membered *meta*-TS is destabilized by 1.4 kcal/mol and the 14-membered *ortho*-TS is destabilized by 3.5 kcal/mol. Visualization of cyclic strain energies (Fig. 6b) was performed for 14,15- and 16-membered palladacycles using StrainViz. This analysis suggests that the *meta*-structure incurs additional strain around the Pd-center, and that in the *ortho*-structure, the biaryl fragment of the directing group experiences additional strain, relative to the more favorable *para*-structure.

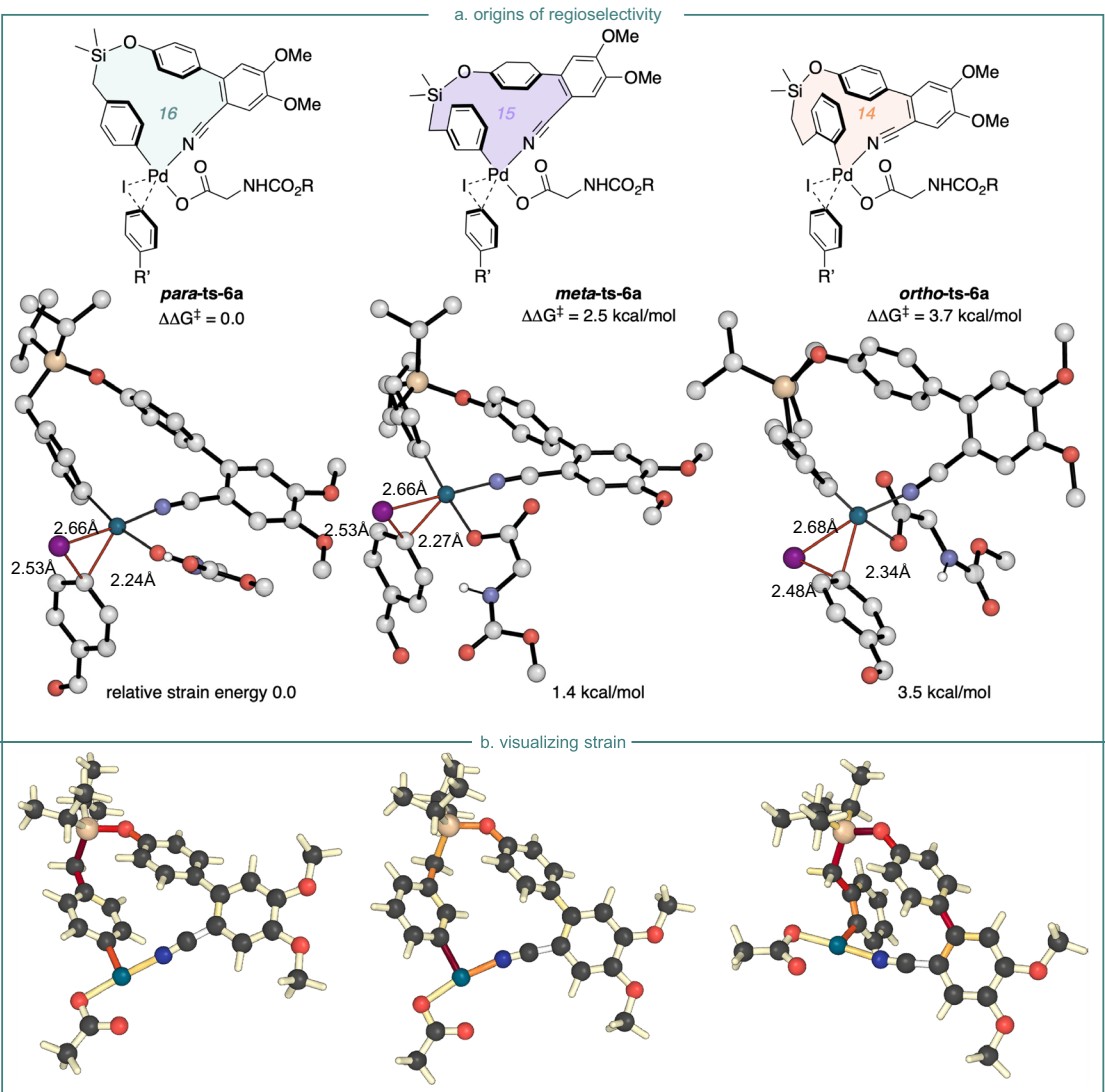

**Fig. 6 Computational analysis of regioselectivity. a** Optimized geometries and corresponding relative Gibbs free energies (kcal/mol) for the *para-*, *meta-*, and *ortho-*TSs in the turnover-determining step. Ring-strain energies are defined by an isodesmic reaction. **b** Visualization of ring strain with StrainViz. Bonds bearing high strain are shown in red.

## Discussion

In summary, we have developed a robust strategy for *para*-C−H arylation employing a removable directing group. The protocol is tolerant of diverse functionalities in arenes and aryl iodides, irrespective of their orientation. Noteworthy, this transformation is amenable in introducing even the heteroarenes selectively at the distal *para*-position, the first report of its kind. Furthermore, aryl iodides containing *O*-glycosides such as mono and di-saccharides were selectively installed at the *para*-position without any hindrance in the catalytic activity. The protocol also paves the path for drug diversification by forming preferred *para*-connected 'flatland' biaryls in various marketed drugs. Post-functionalization removal of the directing group was carried out under mild conditions and further utilized for iterative functionalization of arene with increased aromatic content. The protocol has also been utilized for a concise synthesis of drug molecules. The compatibility of this protocol with diverse functional groups renders the method suitable for streamlining drug development by expanding the chemical space. The reaction mechanism involves relatively facile and reversible C−H activation, followed by turnover-limiting oxidative insertion into the aryl iodide and then reductive elimination to yield the biaryl product. The high levels of regiocontrol for the *para*-product result from the minimization of strain in the macrocyclic template, which is 1.4–3.5 kcal/mol larger in the competing regioisomeric pathways.

## Methods

**Palladium-catalyzed *para*-C−H arylation.** In an oven-dried screw cap reaction tube charged with a magnetic stir bar was added Pd(OAc)₂ (10 mol%), Fmoc-Gly-OH (20 mol%), Ag₂SO₄ (2 equiv.), Cu₂Cr₂O₅ (2 equiv.) and LiOAc.2H₂O (0.5 equiv.). After that, benzylsilyl ether substrate (0.1 mmol, 1 equiv.) and aryl iodide (1 equiv.) were added. Subsequently, 1 mL of 1,1,1,3,3,3-Hexafluoro-2-propanol (HFIP) was added with a disposable laboratory syringe under aerobic conditions. The tube was placed in a preheated oil bath at 80 °C, and the reaction mixture was stirred for 24 h. The reaction mixture was then cooled to room temperature and filtered through a celite pad with ethyl acetate. The filtrate was concentrated in vacuo, and the resulting residue was purified by silica gel (100–200 mesh size) column chromatography to give the desired product.

## Data availability

The authors declare that all other data supporting the findings of this study are available within the article and its Supplementary Information files. The experimental procedures and characterization of all new compounds are provided in Supplementary Information

file. For the computed stationary points and Cartesian coordinates, see Supplementary Data 1 file.

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

## Acknowledgements

Financial support received from SERB; India is gratefully acknowledged (CRG/2018/003951). Financial support received from UGC-India (fellowship to S.M. and T.B.), IIT Bombay (S.G. and S.S.), and (J. C. Bose Fellowship to G.K.L) is gratefully acknowledged. R.S.P. acknowledges support from the NSF (CHE-1955876) and computational resources from the RMACC Summit supercomputer supported by the National Science Foundation (ACI-1532235 and ACI-1532236), the University of Colorado Boulder and Colorado State University, and the Extreme Science and Engineering Discovery Environment (XSEDE) through allocation TG-CHE180056.

## Author contributions

S.M., G.K.L., and D.M. conceived the concept. S.M., S.S., and T.B. performed the reactions, analyzed the products, and carried out the kinetics studies. Y.L. carried out the computational studies. R.S.P. supervised the computational studies. D.M. supervised the experimental work. S.M., S.G., Y.L., G.K.L., R.S.P., and D.M. wrote the manuscript.

## Competing interests

The authors declare no competing interests.
