## [Peer Review File · Nature Communications]

REVIEWER COMMENTS

Reviewer #1 (Remarks to the Author):

The manuscript by Maiti and co-workers reported a para-C-H arylation of benzyl-silane derivatives. Although the arenes were limited to silane derivatives, such directed group assisted para-C-H arylation was not reported before, representing an interesting advance on the remote C-H activation reactions pioneered by Yu group. Considering the importance of biaryl scaffolds in synthetic chemistry, major revisions are required before acceptance for publications:

a) it was claimed in the second paragraph: “the development of a general catalytic system is needed in which (i) the directing group (DG) outcompetes other Lewis basic atoms to bind the catalyst...” However, the substrate scope in Fig.2 or 3 did not contain such structures with N-containing heterocycles. Could the author provide such examples?

b) The major part of reaction conditions, such as CN-based DG design, ligand main structure and HFIP solvent, were discovered by Yu in his seminal work of remote C-H activation, thus this work (Nature 2012, 486 (7404), 518-522.) must be cited after these descriptions: “The para-arylation reaction initiated with Pd(OAc)₂ and NAc-Gly in the presence of AgOAc as oxidant provided the desired product. However, substantial improvement both in yield and selectivity was achieved following further optimization of solvent and additives. Hexafluoroisopropanol (HFIP) solvent, which is beneficial in distal C-H activation, gave superior results to other solvents...”

c) In SI, it was said “Singlet of benzylic proton in ¹H NMR was used to measure the selectivity”, However, from the ¹H NMR, the peaks of benzylic proton from the side products might be overlapped with the products. Thus, at least for the model substrate and those substrates (products 3, 10, 29, 59) whose side products were prominent, it was required to use other deuterated solvents such as d₆-benzene or d₆-acetone to take crude-NMR to see a better separation of the peaks and identification of the selectivity.

d) Explain why in the text of SI: (i) From the NMR of 22, it seems there are two products with bad selectivity. (ii) The NMR of 25 shows 3 products, how could the selectivity be 15:1? (iii) From the NMR of 29, it seems there are two products with bad selectivity.

e) some ¹³C NMR spectra of the products were not good enough, for example the product of 1's peaks were too low.

f) general reaction equation was not shown for figure 2 and 3 and must be added.

g) The industry synthetic route of drug molecule 72 should be presented. And comparison to show the superiority of the new method should be made.

Reviewer #2 (Remarks to the Author):

In this paper, the authors reported a robust strategy for para-C-H arylation employing a removable directing group. The compatibility of this protocol with diverse functional groups renders the method suitable for streamlining drug development by expanding the chemical space. The detailed reaction mechanism was also investigated by means of DFT calculations, which was concluded to be resulted from the minimization of strain in the macrocyclic template. I think that the manuscript could be accepted for publication in Nature Communications after major revision.

-In Figure 5, the structures of ts-3a and ts-3b should be wrong.

-The key bond distances of transition states should be indicated in Figure 6.

-It is known that Pd(OAc)₂ should exist in the trimer form. The authors should re-calculate all the energies to check if the conclusions remain the same.

-In the SI, the energy profile of the Pd–Ag heterodimeric catalyzed pathway was provided. The results show that the C-H bond cleavage and the reductive elimination are lower in energy than those of the PdL monomer model. Therefore the pathway presented in Figure 5 are not the most favorable pathway.

Reviewer #3 (Remarks to the Author):

The research work entitled "Expanding chemical space by para-C₂H arylation of arenes" was presented by Maiti et al. In this article, the author/s have exploited the interesting para-C₂H arylation using their previously established removable directing group (Chem. Sci., 2019, 10, 7426-7432; Angew. Chem., 2016, 128, 7882-7886). So far, only a couple of reports are available in the para-C₂H arylation by Yu and Maiti research groups. The catalytic system is designed exclusively for the para-selective C₂H arylation and differs from their previous para-arylation work. Overall, the chemistry sounds good showed broad substrate scope, including hetero-arylation. Besides, the manuscript has been embedded with some applications on structurally diversified drugs and natural products; the synthesis of an antifungal drug, bifonazole, is impressive. Thus, this reviewer is in favor of suggesting its acceptance in Nature Communications. However, some suggestions/corrections need to be addressed.

1. Follow uniform writing style for the authors' names in the references. For example, in some cases, all the author names are written, whereas in some cases, only the author's name and et al. are included (ref 11, 13, 21, 22, etc.).
2. Check the editor's name in ref 19.
3. The representation of the directing group is not the same in the manuscript and supporting information (Such as DGp and DG1). It would be good to make it uniform in all cases.
4. The author is stating that only the para-arylated product 11 is formed when the reaction is carried out with 4-iodobenzaldehyde and 4-iodobenzyl alcohol individually. However, only the yield and selectivity are mentioned for the aldehyde substrate. Therefore, it is suggested to furnish the yield and selectivity when the reaction is conducted with the 4-iodobenzyl alcohol.
5. What will happen if the second arylation is tried again on 40/41 under these conditions with different iodoarenes?
6. In Figure 4, please give sub-headings for each partition in the ChemDraw file to avoid confusion.
7. Provide numbering for the precursors and final molecules in Tables S1-S12 and the template screening in the supporting information.
8. The ChemDraw part and the descriptive part for the Section 2.3.1 are confusing. Please modify accordingly.
9. Give the numbering for all the aryl iodide coupling partners.
10. Draw the structures uniformly. For instance, the diisopropyl groups in the structures 58, 59, and 60 in supporting information are not the same, which you have followed in the entire manuscript.
11. The NMR spectra of compounds (Ex.: 22/23/29, etc.) have some impurities. Good to submit clean NMR spectra.

12. The compound numbers in the NMR spectra of supporting information files must be in bold format.

Indian Institute of Technology Bombay
Department of Chemistry, Powai, Mumbai, India 400076

Debabrata Maiti
Professor
Associate Editor, *The Journal of Organic Chemistry*
E-mail: dmaiti@chem.iitb.ac.in
http://www.dmaiti.com

Submission of Revised Manuscript (NCOMMS-21-47059-T)

The comments from the Reviewers are encouraging and have helped us to improve the quality of the manuscript. We, therefore, are thankful to the Reviewers for their constructive suggestions. We have now modified/clarified the Reviewers' queries and the modifications are incorporated in the revised manuscript and supporting information.

The point wise responses to the Reviewer's comments are as follows;

Reviewer 1:

The manuscript by Maiti and co-workers reported a *para*-C–H arylation of benzyl-silane derivatives. Although the arenes were limited to silane derivatives, such directed group assisted *para*-C–H arylation was not reported before, representing an interesting advance on the remote C–H activation reactions pioneered by Yu group. Considering the importance of biaryl scaffolds in synthetic chemistry, major revisions are required before acceptance for publications:

Author's response: We are extremely encouraged by the reviewer comments. We have tried to address each and every issue and feel very grateful to the reviewer for assisting us improving the manuscript quality. We sincerely hope our modifications are consistent with the reviewer's comments and the reviewer finds this version acceptable to him.

Reviewer's comment 1:

- a) it was claimed in the second paragraph: "the development of a general catalytic system is needed in which (i) the directing group (DG) outcompetes other Lewis basic atoms to bind the catalyst..." However, the substrate scope in Fig. 2 or 3 did not contain such structures with N-containing heterocycles. Could the author provide such examples?

Author's response: We are sincerely thankful to the reviewer for pointing out this statement from the manuscript. Since we have one example (**3aw**) with *N*-containing heterocycle and two examples (**3au**, **3iy**) with *S*-containing heterocycles in fig 2. and 3, so we made this statement in the manuscript.

Reviewer's comment 2:

- b) The major part of reaction conditions, such as CN-based DG design, ligand main structure and HFIP solvent, were discovered by Yu in his seminal work of remote C–H activation, thus this

work (Nature 2012, 486 (7404), 518-522.) must be cited after these descriptions: “The para-arylation reaction initiated with Pd(OAc)₂ and NAc-Gly in the presence of AgOAc as oxidant provided the desired product. However, substantial improvement both in yield and selectivity was achieved following further optimization of solvent and additives. Hexafluoroisopropanol (HFIP) solvent, which is beneficial in distal C-H activation, gave superior results to other solvents...”

Author's response:

We are sincerely thankful to the reviewer for his/her insightful suggestions. In accordance with the reviewer's comment, we have now cited the above-mentioned work (Nature 2012, 486, 518-522) at reference 40.

Reviewer's comment 3:

- c) In SI, it was said “Singlet of benzylic proton in ¹H NMR was used to measure the selectivity”, However, from the ¹H NMR, the peaks of benzylic proton from the side products might be overlapped with the products. Thus, at least for the model substrate and those substrates (products 3, 10, 29, 59) whose side products were prominent, it was required to use other deuterated solvents such as d₆-benzene or d₆-acetone to take crude-NMR to see a better separation of the peaks and identification of the selectivity.

Author's response:

We are thankful to the reviewer for pointing out this issue with the identification of selectivity. Certainly, this suggestion will improve the quality of our manuscript. We have now provided ¹H NMR of the crude reaction mixture of compounds **3aa**, **3ac**, **3aj**, **3gm**, and **5uk** using both CDCl₃ and d⁶-benzene solvent in the section 2.5 of supporting information.

Reviewer's comment 4:

- d) Explain why in the text of SI: (i) From the NMR of 22, it seems there are two products with bad selectivity. (ii) The NMR of 25 shows 3 products, how could the selectivity be 15:1? (iii) From the NMR of 29, it seems there are two products with bad selectivity.

Author's response: We sincerely thank the reviewer for his/her careful analysis. Accordingly, we have explained in the section 2.5 of supporting information.

Reviewer's comment 5:

- d) some ¹³C NMR spectra of the products were not good enough, for example the product of 1's peaks were too low.

Author's response:

We are sincerely thankful to the reviewer for his/her insightful suggestions. We have now re-provided the NMR spectra for compounds **3aa**, **3ap**, **3aw**, and **3gm**.

Reviewer's comment 6:

- e) general reaction equation was not shown for figure 2 and 3 and must be added.

Author's response:

We are sincerely thankful to the reviewer for his/her insightful suggestions. We have now incorporated general reaction equation for both fig 2 and 3.

Reviewer's comment 7:

- f) The industry synthetic route of drug molecule 72 should be presented. And comparison to show the superiority of the new method should be made.

Author's response:

We are sincerely thankful to the reviewer for his/her insightful suggestions. Correspondingly, we have now showed the industry synthetic route of drug molecule 15 in Fig. 4f. of the revised manuscript.

Reviewer 2:

In this paper, the authors reported a robust strategy for para-C-H arylation employing a removable directing group. The compatibility of this protocol with diverse functional groups renders the method suitable for streamlining drug development by expanding the chemical space. The detailed reaction mechanism was also investigated by means of DFT calculations, which was concluded to be resulted from the minimization of strain in the macrocyclic template. I think that the manuscript could be accepted for publication in Nature Communications after major revision.

Author's response: We are extremely encouraged by the reviewer comments. We have tried to address each and every issue and feel very grateful to the reviewer for assisting us improving the manuscript quality. We sincerely hope our modifications are consistent with the reviewer's comments and the reviewer finds this version acceptable to him.

Reviewer's comment 1:

- a) In Figure 5, the structures of ts-3a and ts-3b should be wrong.
b) The key bond distances of transition states should be indicated in Figure 6.

Author's response:

We are sincerely thankful to the reviewer for his/her insightful suggestions. We have revised these typos in figure 5 and added the key bond distances of transition states in Figure 6.

Reviewer's comment 2:

It is known that Pd(OAc)₂ should exist in the trimer form. The authors should re-calculate all the energies to check if the conclusions remain the same.

Author's response:

We are sincerely thankful to the reviewer for his/her insightful suggestions. The monomeric, dimeric, and trimeric Pd as the active catalyst were considered in C–H activations. In our reaction system, due to greater dissociation energy of the trimeric catalyst (14.7 kcal/mol), the lowest activation energy of the oxidative addition transition state ts-6a is 39.7 kcal/mol. This ruled out trimeric Pd as the active catalyst in this reaction. We have added the dissociation energy of the trimeric catalyst in SI.

The occurrence of hydrolysis is consistent with the known reactivity of Pd₃(OAc)₆ toward ligands such as amines, phosphines and arsines leading to mononuclear complexes in which the acetate groups are monodentate. Therefore, the Pd monomeric as the active catalyst mechanism were presented in our reaction. [1. T. A. Stephenson, S. M. Morehouse, A. R. Powell, J. P. Heffer and G. Wilkinson, J. Chem. Soc., 1965, 3632.; 2. J. F. Knifton, J. Catal., 1979, 60, 27.; 3. Bakhmutov, V. I.; Berry, J. F.; Cotton, F. A.; Ibragimov, S.; Murillo, C. A. Dalton Trans. 2005, 1989]

Reviewer's comment 3:

In the SI, the energy profile of the Pd–Ag heterodimeric catalyzed pathway was provided. The results show that the C–H bond cleavage and the reductive elimination are lower in energy than those of the PdL monomer model. Therefore, the pathway presented in Figure 5 are not the most favorable pathway.

Author's response:

Thanks for the suggestions. The oxidative addition of aryl iodide is identified as rate- and regioselectivity-determining step for the overall reaction. The oxidative addition of aryl iodide occur via transition state with an energetic span of 29.6 kcal/mol in Pd–Ag heterodimeric mechanism, which is 2.2 kcal/mol higher than that of the PdL monomer model. The overall barrier for the Pd–Ag heterodimeric pathway is slightly higher than that of our PdL monomer models, we focus our attention in the manuscript on monomeric pathways only.

In addition, the presence of AgOAc is as oxidant provided the desired product in our reaction. Although AgOAc showed promising results at early stages, extensive optimization studies through systematic alterations of different parameters led to the finding that a combination of Ag₂SO₄, Cu₂Cr₂O₅ and LiOAc.2H₂O, in appropriate proportion, gave the highest yield of para-arylated product in the experiment. Basically, Ag salt is as oxidant rather than catalyst in the reaction.

Reviewer 3:

The research work entitled “Expanding chemical space by para-C–H arylation of arenes” was presented by Maiti et al. In this article, the author/s have exploited the interesting para-C–H arylation using their previously established removable directing group (Chem. Sci., 2019, 10, 7426-7432; Angew. Chem., 2016, 128, 7882-7886). So far, only a couple of reports are available in the para-C–H arylation by Yu and Maiti research groups. The catalytic system is designed exclusively for the para-selective C–H arylation and differs from their previous para-arylation work. Overall, the chemistry sounds good showed broad substrate scope, including hetero-arylation. Besides, the manuscript has been embedded with some applications on structurally diversified drugs and natural products; the synthesis of an antifungal drug, bifonazole, is impressive. Thus, this reviewer is in favor of suggesting its acceptance in Nature Communications. However, some suggestions/corrections need to be addressed.

Author's response: We are extremely encouraged by the reviewer comments. We have tried to address each and every issue and feel very grateful to the reviewer for assisting us improving the manuscript

quality. We sincerely hope our modifications are consistent with the reviewer's comments and the reviewer finds this version acceptable to him.

Reviewer's comment 1:

Follow uniform writing style for the authors' names in the references. For example, in some cases, all the author names are written, whereas in some cases, only the author's name and et al. are included (ref 11, 13, 21, 22, etc.).

Author's response:

We are sincerely thankful to the reviewer for his/her insightful suggestions. In accordance with reviewer comment, we have made necessary changes in the reference section to maintain uniformity.

Reviewer's comment 2:

Check the editor's name in ref 19.

Author's response:

We are sincerely thankful to the reviewer for pointing out this unintentional mistake. We have now corrected it in the revised manuscript.

Reviewer's comment 3:

The representation of the directing group is not the same in the manuscript and supporting information (Such as DG_p and DG₁). It would be good to make it uniform in all cases.

Author's response:

We are sincerely thankful to the reviewer for his/her insightful suggestions. Correspondingly, we have now modified the representation of directing group as DG₁ throughout the whole manuscript and supporting information.

Reviewer's comment 4:

The author is stating that only the para-arylated product 11 is formed when the reaction is carried out with 4-iodobenzaldehyde and 4-iodobenzyl alcohol individually. However, only the yield and selectivity are mentioned for the aldehyde substrate. Therefore, it is suggested to furnish the yield and selectivity when the reaction is conducted with the 4-iodobenzyl alcohol.

Author's response:

We are sincerely thankful to the reviewer for his/her insightful suggestions. We have provided the corresponding yield and selectivity when the reaction is conducted with 4-iodobenzyl alcohol in the revised manuscript.

Reviewer's comment 5:

What will happen if the second arylation is tried again on 40/41 under these conditions with different iodoarenes?

Author's response:

We are sincerely thankful to the reviewer for pointing out this fact. Accordingly, we have tried second arylation using **3ql/3rm** as substrate. But unfortunately, we did not observe any product formation.

Reviewer's comment 6:

In Figure 4, please give sub-headings for each partition in the ChemDraw file to avoid confusion.

Author's response:

We are sincerely thankful to the reviewer for his/her insightful suggestions. We have added sub-headings for each partition in the ChemDraw of Fig. 3.

Reviewer's comment 7:

Provide numbering for the precursors and final molecules in Tables S1-S12 and the template screening in the supporting information.

Author's response:

We are sincerely thankful to the reviewer for his/her insightful suggestions. Correspondingly, we have provided the numbering for the precursors and final molecules in Table S1-S12 and the template screening in the SI.

Reviewer's comment 8:

The ChemDraw part and the descriptive part for the Section 2.3.1 are confusing. Please modify accordingly.

Author's response:

We are sincerely thankful to the reviewer for his/her insightful suggestions. We have now modified it accordingly in the revised SI.

Reviewer's comment 9:

Give the numbering for all the aryl iodide coupling partners.

Author's response:

We are sincerely thankful to the reviewer for his/her insightful suggestions. We have provided the numbering for all the aryl iodide coupling partners and substrates in the revised manuscript.

Reviewer's comment 10:

Draw the structures uniformly. For instance, the diisopropyl groups in the structures 58, 59, and 60 in supporting information are not the same, which you have followed in the entire manuscript.

Author's response:

We are sincerely thankful to the reviewer for his/her insightful suggestions. Accordingly, we have made necessary modification in the SI.

Reviewer's comment 11:

The NMR spectra of compounds (Ex.: 22/23/29, etc.) have some impurities. Good to submit clean NMR spectra.

Author's response:

We are sincerely thankful to the reviewer for his/her insightful suggestions. We have now re-provided the NMR spectra for compounds **3aa**, **3ap**, **3aw**, and **3gm**.

Reviewer's comment 12:

The compound numbers in the NMR spectra of supporting information files must be in bold format.

Author's response:

We are sincerely thankful to the reviewer for his/her insightful suggestions. Accordingly, we have now made the compound numbers in the NMR spectra of SI in bold format.

I hope the Reviewers' will find the responses satisfactory and will recommend for the final acceptance in Nature Communications.

Sincerely,
Deb

REVIEWER COMMENTS

Reviewer #1 (Remarks to the Author):

Major revisions have been made by the author. However, there are still some issues need to be addressed before final decision to be made:

a) why there are 3 spectra for 3gm in section 2.5. Even considering the isomers of the substrate and products, there are still too many peaks (10 peaks from 2.20ppm to 2.44ppm in the second spectrum). Thus, the selectivity is likely not correct. Moreover, how 26:1 was calculated, since no integrations were indicated? I would suggest to use HPLC to determine the selectivity. In addition, since there is a meta-methyl group, it is possible that product is hard to be produced due to steric hindrance. Could the author provide a more convincing proof of the product's structure and selectivity, since it is not possible to determine the product's structure from its ¹H NMR? The methods for obtaining proof can be: i) remove the DG to get identifiable peaks to determine the product's structure. ii) to get an X-ray structure of the product or the product without DG. iii) synthesize the product using other methods and comparing the ¹H-NMR and ¹³C-NMR.

b) Please show the spectrum for 1c in section 2.5 for comparison.

c) From ¹H NMR, it is clear that the isomer for 5uk was not separated, which should be indicated in Fig. 3's footnote. Moreover, from the crude NMR of 5uk, it is hard to believe the yield can be so high as 75% for 5uk, since the crude NMR was messy and the product's peaks were low. Thus, please provide all the mass of isolated product after the percentage yield for all products in the SI.

Reviewer #2 (Remarks to the Author):

The manuscript has been revised properly according to my previous comments. I support the publication of the work in its current form.

Reviewer #3 (Remarks to the Author):

The research work entitled “Expanding chemical space by para-C-H arylation of arenes” was submitted by Maiti and co-workers. The authors have addressed all the concerns raised previously. The responses received for the raised comments are satisfactory. The manuscript is suitable for publication in Nature Communication, after addressing the following comment.

1. Correct the formatting error in the head note of table S2 in supporting information.

Indian Institute of Technology Bombay
Department of Chemistry, Powai, Mumbai, India 400076

Debabrata Maiti
Professor
Associate Editor, *The Journal of Organic Chemistry*
E-mail: dmaiti@chem.iitb.ac.in
<http://www.dmaiti.com>

Submission of Revised Manuscript (NCOMMS-21-47059A)

The comments from the Reviewers are encouraging and have helped us to improve the quality of the manuscript. We, therefore, are thankful to the Reviewers for their constructive suggestions. We have now modified/clarified the Reviewers' queries and the modifications are incorporated in the revised manuscript and supporting information.

The point wise responses to the Reviewer's comments are as follows;

Reviewer 1:

Major revisions have been made by the author. However, there are still some issues need to be addressed before final decision to be made:

Author's response: We are extremely encouraged by the reviewer comments. We have tried to address each and every issue and feel very grateful to the reviewer for assisting us improving the manuscript quality. We sincerely hope our modifications are consistent with the reviewer's comments and the reviewer finds this version acceptable to him/her.

Reviewer's comment 1:

- a) why there are 3 spectra for 3gm in section 2.5. Even considering the isomers of the substrate and products, there are still too many peaks (10 peaks from 2.20ppm to 2.44ppm in the second spectrum). Thus, the selectivity is likely not correct. Moreover, how 26:1 was calculated, since no integrations were indicated? I would suggest to use HPLC to determine the selectivity. In addition, since there is a meta-methyl group, it is possible that product is hard to be produced due to steric hindrance. Could the author provide a more convincing proof of the product's structure and selectivity, since it is not possible to determine the product's structure from its ¹H NMR? The methods for obtaining proof can be: i) remove the DG to get a identifiable peaks to determine the product's structure. ii) to get an X-ray structure of the product or the product without DG. iii) synthesize the product using other methods and comparing the ¹H-NMR and ¹³C-NMR.

Author's response:

We are sincerely thankful to the reviewer for his/her insightful suggestions and pointing out this issue with the identification of selectivity. Certainly, this suggestion will improve the quality of our manuscript. Since both the benzylic as well as methyl peak were splitting further in the ^1H NMR, we have provided the ^1H NMR of isolated compound **3gm**, substrate **1g** in section 2.5 along with crude reaction mixture for better analysis of spectrum. In accordance with the reviewer's suggestions, we performed HPLC but unfortunately both substrate and products peaks are coming at the same position while running separately. Since we did not find any isomeric peak in ^1H NMR of crude reaction mixture, we conclude that the selectivity would be $\geq 20:1$.

To find support for our desired product formation, we cleaved the DG of compound **3gm** and compared the ^1H and ^{13}C NMR of DG cleaved product with authentic compound prepared from cross-coupling method. Since the NMR are same, we can conclude that our protocol is giving *para* selective product. All the NMR spectrum has been provided in the section 2.5 of revised SI.

Reviewer's comment 2:

- b) Please show the spectrum for **1c** in section 2.5 for comparison.

Author's response:

We are sincerely thankful to the reviewer for his/her insightful suggestions. In accordance with the reviewer's comment, we have now provided the spectrum of compound **1c** in section 2.5 of supporting information.

Reviewer's comment 3:

- c) From ^1H NMR, it is clear that the isomer for **5uk** was not separated, which should be indicated in Fig. 3's footnote. Moreover, from the crude NMR of **5uk**, it is hard to believe the yield can be so high as 75% for **5uk**, since the crude NMR was messy and the product's peaks were low. Thus, please provide all the mass of isolated product after the percentage yield for all products in the SI.

Author's response:

We are sincerely thankful to the reviewer for his/her insightful suggestions. Accordingly, we have indicated this point in the Fig. 3's footnote. Also, in accordance with the reviewer's comment, we have now provided the mass of isolated product in the revised SI.

Reviewer 3:

The research work entitled "Expanding chemical space by *para*-C-H arylation of arenes" was submitted by Maiti and co-workers. The authors have addressed all the concerns raised previously. The responses received for the raised comments are satisfactory. The manuscript is suitable for publication in Nature Communication, after addressing the following comment.

Author's response: We are extremely encouraged by the reviewer comments. We have tried to address each and every issue and feel very grateful to the reviewer for assisting us improving the manuscript

quality. We sincerely hope our modifications are consistent with the reviewer's comments and the reviewer finds this version acceptable to him/her.

Reviewer's comment 1:

Correct the formatting error in the head note of table S2 in supporting information.

Author's response: We are sincerely thankful to the reviewer for his/her insightful suggestions. We have corrected the formatting error in the head note of table S2 in the SI.

I hope the Reviewers' will find the responses satisfactory and will recommend for the final acceptance in Nature Communications.

Sincerely,
Deb

REVIEWERS' COMMENTS

Reviewer #1 (Remarks to the Author):

The responses for the raised comments are generally satisfactory, although it is not very likely that HPLC could not separate the products and substrate. Although the major product's structure have been confirmed satisfactorily, I still have reservations about the selectivity of 3gm (>20:1), since the peak pattern was different of the enlarged NMR range (from 2.20 to 2.45 ppm) of pure product and the crude NMR on page S30 even after the peaks coming from the starting material were not considered. So it would be better to remove this example, although it is not mandatory.

Overall, I support the publication of this manuscript without further review.

Indian Institute of Technology Bombay
Department of Chemistry, Powai, Mumbai, India 400076

Debabrata Maiti
Professor
Associate Editor, *The Journal of Organic Chemistry*
E-mail: dmaiti@chem.iitb.ac.in
<http://www.dmaiti.com>

June 07, 2022

Submission of Revised Manuscript (NCOMMS-21-47059B)

The comments from the Reviewers are encouraging and have helped us to improve the quality of the manuscript. We, therefore, are thankful to the Reviewers for their constructive suggestions. We have now modified/clarified the Reviewers' queries and the modifications are incorporated in the revised manuscript and supporting information.

Reviewer 1:

The responses for the raised comments are generally satisfactory, although it is not very likely that HPCL could not separate the products and substrate. Although the major product's structure have been conformed satisfactorily, I still have reservations about the selectivity of 3gm (>20:1), since the peak pattern was different of the enlarge NMR range (from 2.20 to 2.45 ppm) of pure product and the crude NMR on page S30 even after the peaks coming from the starting material were not considered. So it would be better to remove this example, although it is not mandatory.

Overall, I support the publication of this manuscript without further review.

Author's response: We are extremely encouraged by the reviewer comments. We have tried to address each and every issue and feel very grateful to the reviewer for assisting us improving the manuscript quality. We sincerely hope our modifications are consistent with the reviewer's comments and the reviewer finds this version acceptable to him/her. In accordance with the reviewer's suggestion, we have removed the example of 3gm from the revised manuscript and SI.

I hope the Reviewers' will find the responses satisfactory and will recommend for the final acceptance in Nature Communications.

Sincerely,

Deb